# Considering Hemispheric Specialization in Emotional Face Processing: An Eye Tracking Study in Left- and Right-Lateralised Semantic Dementia

**DOI:** 10.3390/brainsci11091195

**Published:** 2021-09-10

**Authors:** Rosalind Hutchings, Romina Palermo, Jessica L. Hazelton, Olivier Piguet, Fiona Kumfor

**Affiliations:** 1Brain & Mind Centre, School of Psychology, The University of Sydney, Sydney, NSW 2050, Australia; rosalind.hutchings@sydney.edu.au (R.H.); jessica.hazelton@sydney.edu.au (J.L.H.); olivier.piguet@sydney.edu.au (O.P.); 2School of Psychological Science, The University of Western Australia, Perth, WA 6009, Australia; romina.palermo@uwa.edu.au

**Keywords:** frontotemporal dementia, social cognition, face perception, emotion recognition

## Abstract

Face processing relies on a network of occipito-temporal and frontal brain regions. Temporal regions are heavily involved in looking at and processing emotional faces; however, the contribution of each hemisphere to this process remains under debate. Semantic dementia (SD) is a rare neurodegenerative brain condition characterized by anterior temporal lobe atrophy, which is either predominantly left- (left-SD) or right-lateralised (right-SD). This syndrome therefore provides a unique lesion model to understand the role of laterality in emotional face processing. Here, we investigated facial scanning patterns in 10 left-SD and 6 right-SD patients, compared to 22 healthy controls. Eye tracking was recorded via a remote EyeLink 1000 system, while participants passively viewed fearful, happy, and neutral faces over 72 trials. Analyses revealed that right-SD patients had more fixations to the eyes than controls in the Fear (*p* = 0.04) condition only. Right-SD patients also showed more fixations to the eyes than left-SD patients in all conditions: Fear (*p* = 0.01), Happy (*p* = 0.008), and Neutral (*p* = 0.04). In contrast, no differences between controls and left-SD patients were observed for any emotion. No group differences were observed for fixations to the mouth, or the whole face. This study is the first to examine patterns of facial scanning in left- versus right- SD, demonstrating more of a focus on the eyes in right-SD. Neuroimaging analyses showed that degradation of the right superior temporal sulcus was associated with increased fixations to the eyes. Together these results suggest that right lateralised brain regions of the face processing network are involved in the ability to efficiently utilise changeable cues from the face.

## 1. Introduction

Face processing plays a central role in day-to-day interactions, with the face providing a range of cues for social communication. These facial cues can be divided into those informed by “invariant” facial features, which remain relatively stable (e.g., shape of the nose, position of the eyes) and those informed by dynamic, “changeable” features (e.g., eye gaze, lip movements, facial expressions) [1]. While invariant features are useful for recognising somebody’s identity, changeable features, including the display of emotional expressions, are arguably more informative for social communication, providing a wealth of moment-to-moment information about an individual’s state of mind.

Of relevance here, right-hemispheric dominance for invariant feature processing, such as identity, is widely accepted [2,3,4,5]; however, whether laterality is relevant for changeable feature processing, such as emotion perception, is less well established. The eyes are a particularly important changeable feature whereby eyebrow position, whites of the eyes, pupil dilation, and gaze direction signal different emotions [6,7], and hence facilitate social interactions as well as threat-detection [8]. Eye tracking studies have shown that healthy adults engage in a typical “facial scanning” pattern, with individuals tending to fixate on regions of the face which portray emotional meaning, namely the eyes and mouth region of the face. This process is modulated by emotion, with greater fixations to the eyes seen for faces expressing fear and anger [9,10,11]. Here, we focused on facial scanning patterns when viewing fear, happy, and neutral expressions. We focused on fear as it is perhaps the most well studied emotion with respect to patterns of facial scanning and we included happiness and neutral expressions as comparison conditions.

The amygdala and superior temporal sulcus are the two regions most commonly associated with processing of information from the eyes [1]. Specifically, the amygdala has been argued to underlie visual attention to the eyes, as part of its broad role in directing attention to emotionally salient information [9,12,13]. This evidence is predominantly based on individuals with amygdala lesions, who show reduced attention to the eyes, with this reduced attention associated with lower emotion recognition performance [9]. Unilateral amygdala damage results in more subtle emotion processing impairments, irrespective of the laterality of damage [13,14]. Indeed, even in individuals with primary visual cortex damage, discrimination of emotional faces can proceed via the right amygdala [15]. The role of the superior temporal sulcus has been demonstrated using functional MRI studies; this region is preferentially activated when attending to eye gaze [16,17] as well as facial expressions [18,19]. Interestingly, while laterality has not been specifically addressed, it appears that the right rather than the left superior temporal sulcus is more frequently activated when attending to eye gaze (vs. identity) [16,19]. Thus, while the literature has identified key regions within the temporal cortex involved in viewing and processing eyes, to our knowledge no study has directly investigated the relative contribution of each hemisphere.

Semantic dementia (SD) is a rare progressive neurodegenerative brain disorder associated with asymmetrical temporal lobe atrophy. Clinically, SD is characterised by a loss of semantic and conceptual knowledge irrespective of testing modality [20]. This clinical profile reflects underlying atrophy of the anterior temporal lobe, superior temporal sulcus, amygdala, and insular cortex, which is largely left-lateralised [21,22] (hereafter referred to as left-SD). A small proportion (~30%) of patients, however, present with right-lateralised atrophy (hereafter, referred to as right-SD) [23,24,25]. Behavioural symptoms are more pronounced in right-SD and can include disinhibition, depression, and aggressive behaviour [23], as well as significant emotion recognition deficits [26,27] and prosopagnosia [23,27,28]. Right-SD patients also demonstrate more extensive atrophy across the right temporal and frontal lobes, compared to left-SD patients [23,24,29].

To date, differences in the profile of face processing deficits in the two variants of SD remain unclear [30]. Deficits in facial emotion processing have been demonstrated in left-SD [24,31,32,33,34,35,36,37] and right-SD [24,26,27]. A growing number of studies have demonstrated that right-SD patients have worse deficits in facial expression processing, than left-SD [24,27,38,39], although one study reported no difference between groups [40].

In left-SD, impaired expression recognition has been argued to result at least partially from deficits in semantic labelling, due to degradation in the anterior temporal lobe (e.g., [28,41,42]). Evidence from tasks that require participants to discriminate facial expressions (i.e., removing the labelling requirement), however, suggests that the deficit in both left-SD and right-SD is not solely due to a semantic labelling breakdown [24,33]. In right-SD, neural correlates of facial emotion recognition [26] overlap with those regions associated with attention to changeable features [16,43], including the lateral and medial temporal lobe. Notably, whether distinct mechanisms underlie expression processing deficits in left-SD and right-SD has not yet been determined.

Here, we aimed to: (i) establish the degree of overt deficits in facial expression discrimination and facial expression recognition in left-SD and in right-SD, compared with healthy controls, (ii) compare facial scanning and emotion recognition patterns in these groups, and (iii) identify the neural correlates of facial scanning patterns. Given the right-hemisphere dominance in processing facial changeable features suggested by the literature, right-SD patients were predicted to show reduced fixations to the eyes compared to controls, as well as significant deficits on the facial expression discrimination and facial expression recognition tasks. Facial scanning patterns in left-SD were not predicted to differ from controls. With regards to overt performance, it was hypothesised that left-SD would show deficits on the expression discrimination task but that this would be less severe than right-SD patients. Similar deficits in left-SD and right-SD were predicted for expression recognition, given the semantic labelling requirements of this task. Number of fixations to regions of interest was predicted to correlate with atrophy in the right superior temporal sulcus and right amygdala.

## 2. Materials and Methods

### 2.1. Participants

Sixteen SD patients (10 left-SD, 6 right-SD) and 22 healthy control participants were recruited for this study. Patients and healthy control participants were recruited through FRONTIER, the younger-onset dementia research clinic, based in Sydney, Australia. Healthy older control participants were volunteers from the community, or friends and family of the patients. Patient diagnosis was made by an experienced neurologist in consultation with a neuropsychologist and occupational therapist. Diagnosis was based on clinical examination, detailed family history, cognitive assessment, and brain neuroimaging. SD patients were subclassified as either left-SD or right-SD based on the side of predominant temporal lobe atrophy. Individuals classified as left-SD presented with progressive loss of conceptual knowledge, manifesting as anomia, impaired confrontation naming and single-word comprehension deficits, and atrophy predominantly in the left anterior temporal lobe. Patients classified as right-SD presented with prosopagnosia and changes in behaviour, as well as reduced conceptual knowledge ([26,44], see [23]). Patterns of brain atrophy in these patients were typical of SD, but with more marked atrophy on the right than left anterior temporal lobe. Individuals with a significant history of psychiatric or neurological condition, or substance abuse were excluded. All participants were required to have sufficient proficiency in English in order to complete all the tasks and have a minimum of a primary school-level of education (i.e., 6 years). In addition, control participants were required to score ≥ 88/100 on the Addenbrooke’s Cognitive Examination-R (ACE-R) or the Addenbrooke’s Cognitive Examination-III (ACE-III), a general measure of cognition.

### 2.2. Cognitive Assessment

All participants were assessed on the Australian-version of the ACE-R [45] or ACE-III [32], which assesses five cognitive domains including attention/orientation, memory, fluency, language, and visuospatial function. ACE scores were converted using the accepted formula to ensure comparability of scores across versions [46]. Cognitive profiles in left-SD, right-SD, and controls, were determined using subscales of the ACE, as well as additional tests of language (The Sydney Language Battery, SYDBAT, [47]) and visuospatial ability (Rey Complex Figure—Copy, [48]).

### 2.3. Facial Expression Processing Tasks

Participants were assessed on the facial affect discrimination task and facial affect selection task [33,49]. In the facial affect discrimination task, participants were shown pairs of faces and asked to indicate whether the two faces displayed the same facial emotional expression. In the facial affect selection task, participants were shown an array of seven facial expressions (happy, angry, sad, surprise, fear, disgust, and neutral) and were required to point to the face that expressed the emotion cued by the experimenter (e.g., “point to the *angry* face”). All face stimuli are from the NimStim database (http://www.macbrain.org/resources, accessed on 9 September 2021). Images are cropped to remove extraneous features such as hair, and identities are unfamiliar to the participant.

Each task included 42 trials. Performance was untimed and no feedback was given. One point was given for a correct answer and zero for an incorrect answer. Accuracy was converted to a percentage score for statistical analysis.

### 2.4. Eye-Tracking Paradigm

The eye tracking task is the same as reported in [50]. Participants passively viewed faces appearing on a screen and no explicit response was required. The experimenter instructed participants to “look at the face”, with no other instruction given. For each trial, a face appeared for 3000 ms. A light grey ellipse, which approximately matched the face stimuli dimensions appeared in the centre of the screen between trials. Instead of a fixation cross, the oval flashed a lighter shade of grey to cue the next face so that the individual oriented to the space where the face would appear but was not cued to any specific location on the face (Figure 1).

The stimulus set included images from the Pictures of Facial Affect series [51]. Eight individual faces, displaying fear, happy, and neutral were used for this study (i.e., 24 images). These stimuli were repeated across three blocks, resulting in 72 trials in total (i.e., 24 images × 3 blocks). This paradigm was adapted from a task designed for functional MRI research; as such, all images of the same emotion were presented together, with each emotion-set presented in a pseudo-randomised order across blocks.

An LCD monitor was used for stimulus presentation, with screen resolution 1680 × 1050 and 59 Hz refresh rate. The task was programmed in Presentation (Neurobehavioural Systems, Inc., Albany, CA, USA, www.neurobs.com, accessed on 9 September 2021). Stimuli were presented in grey-scale on a grey background. Images were approximately 455 × 655 pixels.

Monocular eye tracking was recorded from the right eye using the EyeLink 1000 (SR Research, Mississauga, ON, Canada), with a sampling rate of 500 Hz. Binocular recording was not deemed necessary because the stimuli were presented at the same distance, and we were not measuring vergence or microsaccades. We assumed that eye movements in all participants were conjugate. A remote camera mount was used, positioned directly under the stimulus presentation screen. Participants were seated 55–60 cm away from the stimulus screen. A five-point calibration and validation procedure were used prior to starting the task, with gaze position error typically less than 1°. A buffer period of 500 ms occurred at the end of each trial, where no eye tracking data were recorded. Therefore, eye movement data were recorded for the first 2500 ms that the face stimulus was on the screen. For analysis, three regions of interest were defined; (i) the whole face, (ii) the eyes, and (iii) the mouth (see Figure 2).

The primary measure of interest was number of fixations. Given the aim of this study was to assess the role of laterality in processing changeable facial cues, analyses focused on fixations to the eyes and the mouth. Prior to analysing fixations to these two regions of interest, fixations to the whole face including the eyes and mouth, were measured to ensure this was not driving any group differences.

### 2.5. Statistical Analyses

A chi-square test was used to assess group differences in the distribution of sex. ANOVAs were used to investigate differences on continuous variables. A one-way ANOVA was used to assess group differences in the number and duration of fixations to the whole face, followed by repeated measures ANOVAs to investigate group differences for each region of interest (eyes and mouth), with group (controls, left-SD, right-SD) as the between-subjects variable and emotion (fear, happy, neutral) as the within-subjects variable. Significant effects were followed up with *post hoc* pairwise comparisons, using Sidak correction for multiple comparisons. Effect sizes are reported using partial eta-squared (η_p_^2^) where relevant. The relationship between number of fixations to the eyes and expression recognition was conducted using one-tailed Pearson’s correlation.

### 2.6. Voxel-Based Morphometry (VBM) Procedure

Whole-brain structural Magnetic Resonance Imaging (MRI) scans were obtained using a 3-Tesla scanner. High resolution T1-images were acquired using the following sequences: coronal orientation, matrix 256 × 256, 200 slices, 1 × 1 mm in-plane resolution, slice thickness 1 mm, echo time/repetition time = 2.6/5.8 ms, flip angle α = 8°. MRI scans were required to: (i) be of sufficient quality for neuroimaging analysis, (ii) acquired on the same scanner, and (iii) for SD patients to be acquired within 6 months of the behavioural task. Thirty-two scans (20 controls, 7 left-SD, 5 right-SD) were available for VBM analysis using FSL (FMRIB Software Library, Oxford, UK, https://fsl.fmrib.ox.ac.uk, accessed on 9 September 2021). One scan was acquired on a different MRI scanner, two individuals did not undergo scanning due to MRI contraindications, two were excluded due to poor quality, and one control was excluded due to an incidental finding on MRI. Structural images were brain-extracted using BET, then tissue segmentation was conducted with automatic segmentation (FAST) [52]. Grey matter partial volume maps were aligned to Montreal Neurological Institute (MNI) standard space (MNI152) using non-linear registration (FNIRT) [53] which uses a *b*-spline representation of the registration warp field [54].

A study-specific template was created, and the native grey matter images were non-linearly re-registered. Modulation of the registered partial volume maps was carried out by dividing them by the Jacobian of the warp field, and the modulated, segmented images were smoothed with an isotropic Gaussian kernel with a sigma of 3 mm.

A voxel-wise GLM was applied to investigate grey matter intensity differences between (i) each patient group and controls, and (ii) between patient groups, using *t*-tests with permutation-based, non-parametric tests, with 5000 permutations per contrast (Nichols and Holmes, 2001).

To examine the neural correlates of fixation patterns, number of fixations was entered into a GLM that included all participants, to achieve greater variance in scores [55,56]. The statistical threshold was set at *p* < 0.005, uncorrected for multiple comparisons, with a conservative cluster extent threshold of 150 voxels. This statistical threshold was selected to balance the risk of Type I and Type II error [57,58]. Anatomical locations of significant results were overlaid on the MNI standard brain, with maximum coordinates provided in MNI stereotaxic space. Anatomical labels were determined with reference to the Harvard-Oxford probabilistic cortical and subcortical atlases.

## 3. Results

### 3.1. Demographics and Neuropsychological Performance

Participant demographic and cognitive performance is shown in Table 1. No group differences were observed for age (*p* = 0.706), education (*p* = 0.631) or sex (*p* = 0.721). Patient groups did not differ in disease duration (*p* = 0.699) or in disease severity (*p* = 0.679).

A significant effect of group was evident on the ACE (*p* < 0.001), with both patient groups performing worse than controls (left-SD, *p* < 0.001; right-SD, *p* = 0.002), but no difference between left-SD and right-SD (*p* = 0.301). Deficits on the language subdomain of the ACE as well as the SYDBAT were evident in both the left-SD and right-SD groups (all *p* values < 0.001). No group difference was observed on the visuospatial subdomain of the ACE (*p* = 0.110), although right-SD showed impaired performance on the Rey Complex Figure Copy (*p* = 0.003). Both SD groups were impaired on fluency (both *p* values < 0.001), memory (left-SD, *p* < 0.001; right-SD, *p* = 0.010) and language (both *p* values < 0.001) compared to controls. Left-SD also performed worse on the orientation/attention subscale compared to controls (*p* = 0.029). Patient groups did not differ on any neuropsychological measure (all *p* values > 0.15).

### 3.2. Facial Expression Recognition Performance

Performance on the facial expression tasks is displayed in Table 2. Group differences were evident on both the facial affect discrimination task (*F*(2,34) = 10.441, *p* < 0.001) and facial affect selection task (*F*(2,34) = 23.103, *p* < 0.001). Post hoc comparisons indicated that both patient groups performed worse than controls on the facial affect discrimination task (left-SD, *p* = 0.011; right-SD *p* = 0.010) and the facial affect selection task (left-SD, *p* = 0.002; right-SD, *p* < 0.001). Patient groups did not differ in facial affect discrimination performance (*p* = 0.573); however, right-SD patients were significantly impaired compared to left-SD on the facial affect selection task (*p* = 0.021).

Next, performance on trials where participants were asked to identify fear, happy or neutral faces on the facial affect selection task was assessed (given that these were the expressions viewed in the eye-tracking task). Group differences were evident for all emotions (fear (*F*(2,34) = 12.092, *p* < 0.001); happy (*F*(2,34) = 3.457, *p* = 0.043); neutral (*F*(2,34) = 18.788, *p <* 0.001)). Compared with controls, both left-SD and right-SD showed deficits in fear recognition (left-SD, *p* = 0.029; right-SD, *p* = 0.001), but only right-SD were impaired for happy (*p* = 0.043) and neutral (*p* < 0.001) emotions. Right-SD also showed significant deficits compared to left-SD for neutral faces (*p* < 0.001), but not for fear (*p* = 0.153) or happy (*p* = 0.108) faces.

### 3.3. Eye-Tracking Analyses

#### 3.3.1. Number of Fixations

First, number of fixations was examined. In total, three outliers were identified, one for each region of interest. All three outliers were left-SD participants and had fixations greater than two standard deviations away from their group mean. Data for these participants were therefore removed from the relevant analysis. Number of fixations to the whole face did not significantly differ between groups (*F*(2,34) = 1.515, *p* = 0.234). Thus, any effect of group observed for the two key regions of interest (eyes and mouth) was not considered to reflect differences in fixations to the whole face.

Average number of fixations to the eyes and mouth are presented in Figure 3. For fixations to the eyes, analyses revealed a significant main effect of group (*F*(2,34) = 4.725, *p* = 0.015, η_p_^2^ = 0.217). Overall, right-SD had significantly greater fixations to the eyes than left-SD (*p* = 0.013) and marginally more fixations to the eyes than controls (*p* = 0.079). Differences between left-SD and right-SD were observed in all three emotion conditions (fear, *p* = 0.017; happy, *p* = 0.010; neutral *p* = 0.023). While the mean number of fixations was lower in left-SD than controls, this difference was not statistically significant averaged across emotions (*p* = 0.436), or for any individual emotional condition (all *p* values > 0.7).

A main effect of emotion was also evident for number of fixations to the eyes (*F*(2,68) = 35.514, *p* < 0.001, η_p_^2^ = 0.511). Here, participants showed the fewest fixations to the eyes in the happy condition, compared to both the fear and neutral conditions (both *p* values < 0.001) with no difference between the fear and neutral conditions (*p* = 0.524). No interaction, however, was evident between group and emotion (*F*(4,68) = 0.471, *p* = 0.757, η_p_^2^ = 0.027), suggesting that participants modulated their scanning patterns in a similar way across emotions.

For fixations to the mouth, no significant main effect of group was evident (*F*(2,34) = 0.027, *p* = 0.973, η_p_^2^ = 0.002). The main effect of emotion was significant (*F*(2,68) = 11.106, *p* < 0.001, η_p_^2^ = 0.246), with more fixations to the mouth observed in the happy condition than both the fear (*p* = 0.025) and neutral (*p* = 0.001) conditions. Again, no interaction between group and emotion was evident (*F*(4,68) = 0.286, *p* = 0.886, η_p_^2^ = 0.017).

Figure 3C shows count-based fixation heatmaps depicting patterns of facial scanning in each group averaged across emotion conditions. Here, areas of the face highlighted in red represent where more fixations occurred based on the maximum per-trial-average fixation count averaged across all trials. Complementary to our statistical analyses, the heat map for right-SD highlights the eye region more compared to the heat maps for healthy controls and left-SD.

#### 3.3.2. Relationship between Fixations and Facial Expression Recognition

Finally, correlations between facial affect selection performance and number of fixations to the eyes in each group were explored. The focus was the fear condition, given the role the eye region plays in the recognition of negative expressions (see Figure 4). The correlation between number of fixations to the eyes and facial affect selection was not significant in controls (*r* = 0.221, *p* = 0.323). Left-SD showed a moderate association, but it did not reach statistical significance (*r* = 0.479, *p* = 0.229). In right-SD, a large and significant correlation (*r* = 0.857, *p* = 0.029) was observed. In both left-SD and right-SD, a greater number of fixations to the eyes was associated with better fear recognition.

### 3.4. Neuroimaging Acquisition and Analysis

#### 3.4.1. Atrophy Analysis

Figure 5 displays regional changes in brain integrity in patient groups compared with controls (Figure 5A) and between patient groups (Figure 5B). Compared with controls, left-SD displayed reduced grey matter integrity in bilateral but asymmetric temporal regions, which predominantly affected the left hemisphere as expected. This included the left temporal pole and insula, as well as the left amygdala and hippocampus. In contrast, reduced intensity in right-SD included widespread bilateral but asymmetric temporal regions predominantly in the right hemisphere, including the right inferior temporal gyrus, right fusiform cortex, right temporal pole, right insula, as well as the right amygdala and hippocampus. Between patient group comparisons showed more extensive reduction in grey matter integrity in right-SD than in left-SD, in the right posterior temporal cortex. The converse contrast showed a small region of lower grey matter integrity in the left frontal pole in left-SD than in right-SD.

#### 3.4.2. Covariate Analysis: Fixations to the Eyes

Next, the relationship between grey matter intensity and number of fixations to the eyes was investigated in all participants combined, averaged across emotions. Regions associated with fixations are depicted in Figure 6 and reported in Table 3. Increased fixations were associated with reduced grey matter intensity in the right temporal regions including the planum temporale and superior temporal gyrus. Analysis of individual emotion categories showed similar results (see Appendix A).

## 4. Discussion

The current study aimed to tease apart the role of laterality in processing changeable facial cues, specifically emotion, in SD patients with predominantly left- or right-lateralised temporal atrophy. Results revealed that the right and left hemisphere play distinct roles in patterns of facial scanning, with right-SD patients showing more fixations to the eyes than both left-SD and controls, when viewing facial expressions. This pattern was evident across all three emotional conditions (fear, happy, neutral). In contrast, left-SD patients showed a similar pattern of fixations to controls. Fixations to the mouth did not differ across groups. Interestingly, patterns of fixations were sensitive to emotional content in all groups (i.e., more fixations to the eyes when viewing fear, relative to happy facial expressions). Despite looking at key regions of the face, both patient groups showed deficits in facial affect discrimination and selection.

Both healthy controls and left-SD patients showed a typical triangular pattern of facial scanning, which centred on the eyes and mouth region. In contrast, while right-SD also attended to these regions of the face, their facial scanning pattern was more diffuse, and they showed significantly more fixations to the eyes than both controls and left-SD. The facial scanning patterns observed in right-SD provide evidence for the contribution of the right hemisphere in processing facial cues, expanding the existing literature investigating both invariant [2,3] and changeable feature processing [16,19,59,60,61]. In particular, our VBM analyses demonstrated that *increased* number of fixations was associated with reduced integrity of the right superior temporal sulcus. These results suggest that degradation of the right superior temporal sulcus reduces the ability to efficiently process changeable cues in right-SD. Indeed, right-SD patients showed increased fixations to the eyes across all emotional expressions, suggesting that right-hemispheric damage creates a widespread disruption in processing facial features, rather than an emotion-specific deficit. The superior temporal sulcus is known to be important for processing changeable facial cues [16,17,19,59,61,62,63]. Notably, an fMRI study showed lateralisation to the right hemisphere when processing dynamic faces [61]. This lateralisation was seen in human but not monkeys and was interpreted as evidence that the emergence of language function in humans contributes to the lateralisation of face processing [61]. Dynamic faces may be more naturalistic than static faces, and hence comparison between studies using dynamic and static faces should be done with caution. Future studies using dynamic stimuli in left- and right-lateralised SD will help to provide converging evidence of the critical role of the right hemisphere for processing both static and dynamic face stimuli. It should also be noted that some studies have reported that healthy adults show a left-sided bias with the distribution of the first fixation tending to be just to the left of the centre of the nose [64]. While we did not investigate first fixations, our heatmaps show no clear laterality effects in any of our groups, when averaged across the duration of the trial. Nonetheless, future analyses examining first fixations may be useful to extrapolate other potential differences between right-SD, left-SD, and healthy controls.

Differences between left-SD and right-SD patients were less clear-cut with regard to overt emotion perception performance. Contrary to predictions, both SD groups showed similar levels of deficit in facial affect discrimination, suggesting that a breakdown in both left-SD and right-SD goes beyond semantic labelling. As expected, both patient groups showed deficits on facial affect selection. Right-SD patients, however, showed impaired recognition of more emotion categories (happy and neutral expressions, as well as fear) than left-SD patients. While previous studies have demonstrated that right-SD patients have more impaired expression labelling than left-SD [24,27,38], studies rarely report this performance by emotion category. The results of the current study therefore build on the previous literature, suggesting that the severe deficits reported in right-SD may be due to a widespread breakdown across emotions in this group, that is not evident in left-SD.

Interestingly, the correlation between fixation patterns and overt expression recognition suggested that a greater number of fixations to the eyes was beneficial for emotion recognition in right-SD. Such evidence complements the findings of Adolphs and colleagues [9], who demonstrated improved expression recognition in an amygdala lesion patient when visual attention was redirected to the eyes of emotional faces. Notably, the profile of performance in SD stands in contrast to what we previously observed in bvFTD, where increased fixations to the eyes was not associated with performance on tasks of emotion perception [50]. While it is tempting to consider the potential functional impact of increased fixations in right-SD, this correlation should be interpreted with caution due to the relatively small sample of participants included in the analysis. Indeed, whether increased looking at emotional regions of the face leads to functional improvement in performance is somewhat variable. Here, we did not see a relationship between number of fixations and emotion perception in healthy older adults, although we have observed this relationship previously [65]. Nevertheless, these preliminary findings provide impetus for future studies examining the relationship between fixation patterns and overt expression recognition both in healthy adults and patients, and provide a potential avenue for intervention studies aimed at enhancing emotion processing, particularly in right-SD.

It should be noted that our findings reflect the clinical profile which is observed relatively early in the disease stage. With disease progression, atrophy in left- and right-SD encroaches into the contralateral hemispheres, as well as regions beyond the temporal lobe including medial prefrontal regions, which leads to the clinical profiles of these syndromes becoming more similar over time [24,66]. Interventions such as directing attention to key regions of the face are likely to be more effective early in the disease course. In addition to patterns of facial scanning, pupil responses also change in response to emotional stimuli, presumably reflecting changes in autonomic arousal [67]. While abnormalities in autonomic arousal have received increasing attention in frontotemporal dementia particularly with respect to emotion processing [68,69,70], to our knowledge, pupil responses to emotional facial expressions have not been investigated and represents an interesting avenue for future studies. One of the limitations of this study is that we only focused on the emotions of fear and happiness. While patients with SD show pervasive emotion processing deficits, individual emotions may not be equally affected. Moreover, patterns of facial scanning vary according to emotion, with the relevant contribution of the mouth region and the eye region different depending on the emotion [71,72]. Future studies that consider how other emotions are affected are warranted.

## 5. Conclusions

In summary, our findings expand on studies in healthy adults and demonstrate that the right hemisphere is critical for processing facial expressions of emotion, even on tasks where decoding of emotion is not explicitly required. While existing models of face processing, such as proposed by Haxby [1] do not explicitly consider laterality, updated theoretical models of face processing should incorporate the specialization of the right hemisphere for not only identity but also processing of emotion.

## Figures and Tables

**Figure 1 brainsci-11-01195-f001:**
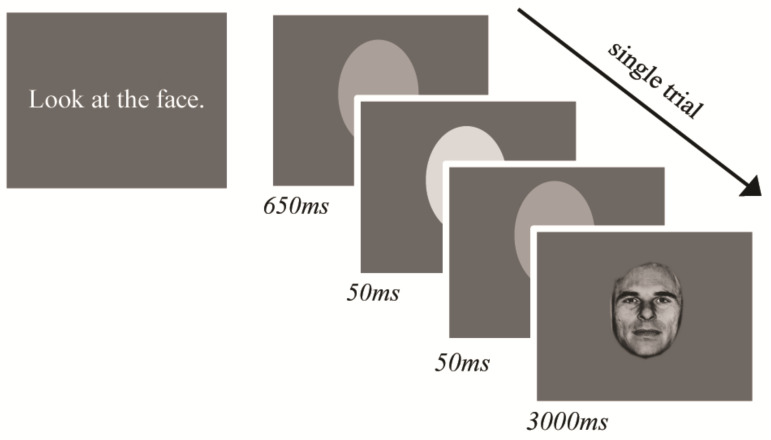
Eye-tracking paradigm trial structure and stimulus timing.

**Figure 2 brainsci-11-01195-f002:**
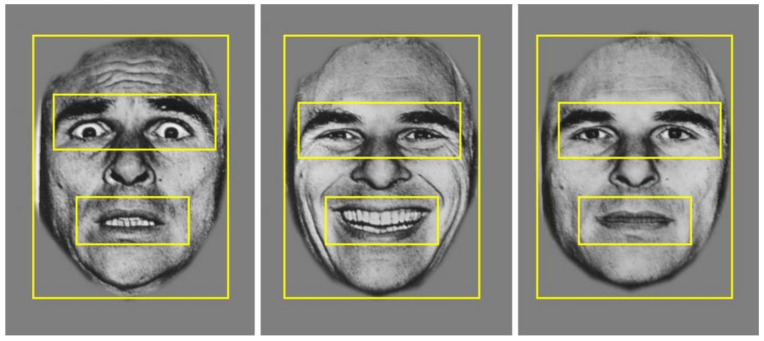
Example of the three regions of interest: the whole face, the eyes, and the mouth, applied across stimuli for data analysis.

**Figure 3 brainsci-11-01195-f003:**
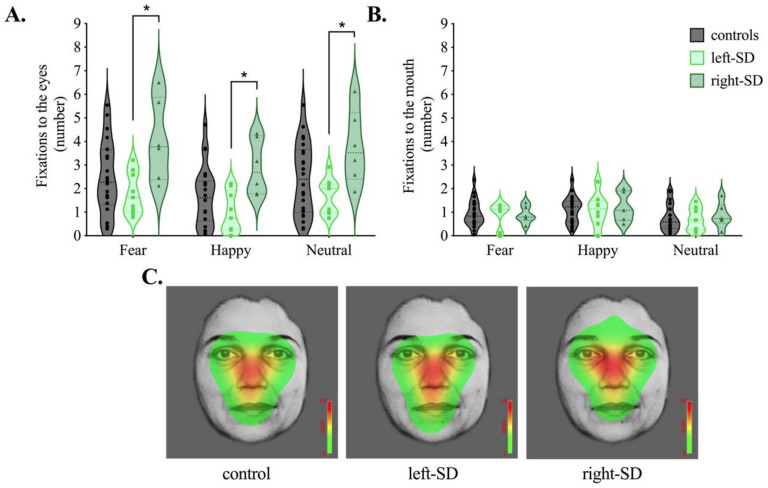
Average number of fixations to the (**A**) eyes and (**B**) mouth in controls (grey), left-lateralised semantic dementia (left-SD, light green) and right-lateralised semantic dementia (right-SD, dark green) for fear, happy and neutral conditions. (**C**) Heat maps corresponding to the maximum per-trial average for the number of fixations across conditions (fear, happy, neutral) in each group. Colours on the heat map represent how often participants looked at the faces on average, with red representing areas where the most fixations occurred. * *p* < 0.05.

**Figure 4 brainsci-11-01195-f004:**
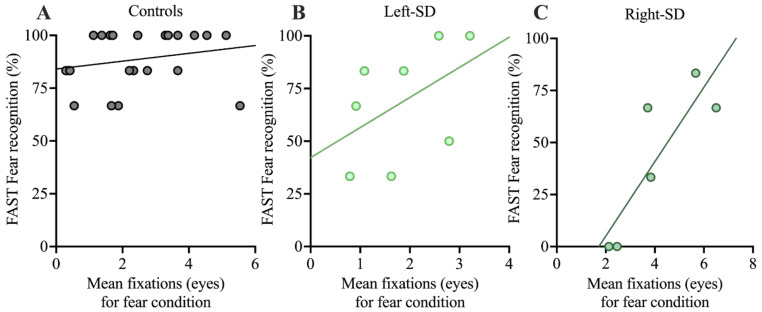
Correlations between performance on the FAST for fear recognition and mean number of fixations to the eyes in the fearful condition in **A:** controls, **B**: left-SD, and **C**: right-SD. FAST: Facial Affect Selection Task.

**Figure 5 brainsci-11-01195-f005:**
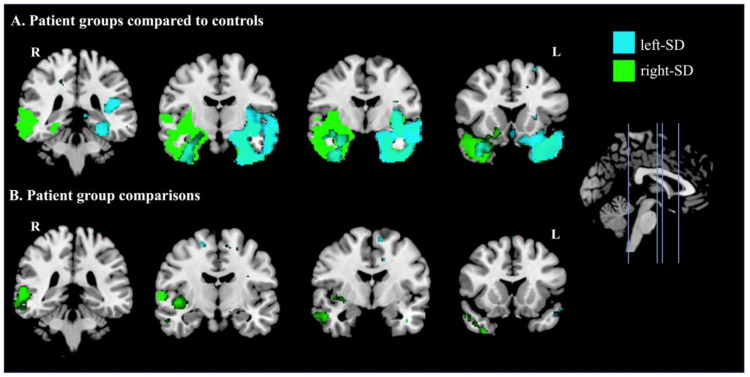
Patterns of atrophy in **A**: left-lateralised semantic dementia (left-SD) compared to controls (cyan) and right-lateralised semantic dementia (right-SD) compared to controls (green); and **B:** where left-SD showed lower intensity than right-SD (cyan) and where right-SD showed lower intensity than left-SD (green). Coloured voxels show regions that were significant in a voxel-wise analysis at *p* < 0.005, uncorrected, with a cluster extent threshold of 150 voxels. MNI coordinates: *x* = 0; *y* = −36, −10, −4, 16 (left to right). L = left; R = right.

**Figure 6 brainsci-11-01195-f006:**
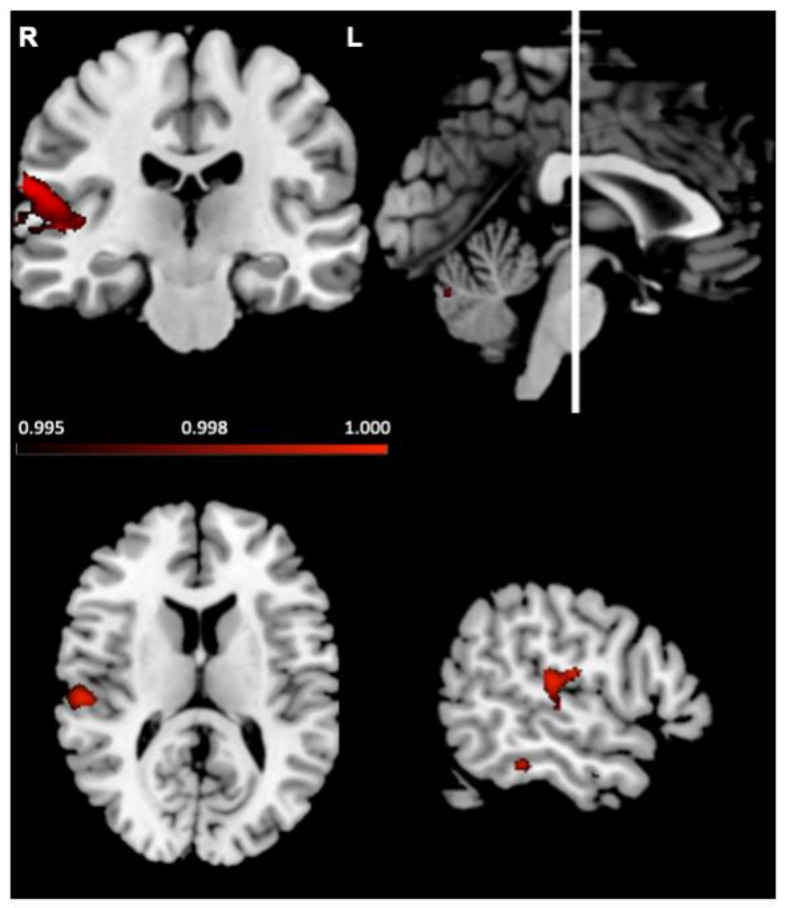
Voxel-based morphometry analyses, in left-lateralised semantic dementia, right-lateralised semantic dementia and controls combined, showing brain regions where reduced grey matter intensity correlated with more fixations to the eyes. Coloured voxels show regions that were significant in a voxel-wise analysis at *p* < 0.005, uncorrected, with a cluster extent threshold of 150 voxels. R = right; L = left. MNI coordinates: x = 55, y = −22, z = 13.

**Table 1 brainsci-11-01195-t001:** Demographic information and neuropsychological performance in semantic dementia patients and controls.

	Controls (*n* = 22)	Left-SD (*n* = 10)	Right-SD (*n* = 6)	F Value	*p* Value	Post Hoc
Age (years)	65.0 ± 7.1	65.6 ± 5.1	62.6 ± 10.3	0.4	ns	-
Education (years)	14.0 ± 2.5	12.9 ± 3.4	13.5 ± 3.8	0.5	ns	-
Sex (m/f)	11/11	6/4	4/2	0.7 ^	ns	-
Disease Duration (months)	n/a	75.3 ± 30.7	81.9 ± 35.6	0.7	ns	-
FRS ^a^ (Rasch Score)	n/a	0.04 ± 1.7	0.4 ± 0.9	0.2	ns	-
SYDBAT—Naming ^b^ (/30)	27.5 ± 1.5	9.6 ± 5.1	13.0 ± 10.1	54.5	**	left-SD, right-SD < controls
SYDBAT—Comprehension ^b^ (/30)	29.5 ± 0.9	20.0 ± 7.6	18.0 ± 10.1	16.9	**	left-SD, right-SD < controls
RCF—Copy (/36)	33.1 ± 2.1	31.4 ± 3.1	29.1 ± 2.8	6.8	*	right-SD < controls
ACE (/100)	95.8 ± 2.4	59.8 ± 22.2	71.7 ± 22.1	24.5	**	left-SD, right-SD < controls
ACE subdomains (%)						
Fluency	88.3 ± 9.0	37.9 ± 25.7	50.0 ± 32.6	27.2	**	left-SD, right-SD < controls
Orientation/Attention	96.2 ± 6.3	80.6 ± 25.7	89.8 ± 15.1	3.8	*	left-SD < controls
Memory	94.8 ± 5.9	51.5 ± 30.0	69.2 ± 27.1	21.7	**	left-SD, right-SD < controls
Language	99.0 ± 2.1	48.8 ± 21.3	63.5 ± 26.2	44.3	**	left-SD, right-SD < controls
Visuospatial	98.6 ± 2.7	86.9 ± 27.9	87.5 ± 19.8	2.3	ns	-

Note: Values are mean ± standard deviation. ^ χ^2^ value; Where relevant, parentheses indicate the maximum possible score. left-SD = left-lateralised semantic dementia; right-SD = right-lateralised semantic dementia; ACE-III = Addenbrooke’s Cognitive Examination-III; RCF = Rey Complex Figure; SYDBAT = Sydney Language Battery; ns = *p* > 0.05; * *p* < 0.05; ** *p* ≤ 0.001. Missing data: ^a^ 1 left-SD; ^b^ 2 left-SD, 1 right-SD.

**Table 2 brainsci-11-01195-t002:** Performance on facial affect tasks in semantic dementia patients and controls.

	Controls (n = 22)	Left-SD (n = 10)	Right-SD (n = 6)	F Value	*p* Value	*Post Hoc*
Facial affect discrimination (%)	87.2 ± 3.1	79.9 ± 8.6	76.2 ± 9.2	10.4	**	left-SD, right-SD < controls
Facial affect selection (%)	92.4 ± 5.8	78.0 ± 8.8	63.1 ± 20.0	23.1	**	right-SD < left-SD < controls
Facial affect selection task by emotion						
Fear (%)	88.6 ± 13.0	64.8 ± 28.2	41.7 ± 36.1	12.5	**	right-SD, left-SD < controls
Happy (%)	98.5 ± 4.9	98.1± 5.6	88.9 ± 17.2	3.0	*	right-SD < controls
Neutral (%)	100.0 ± 0.0	98.1 ± 5.6	72.2 ± 25.1	24.1	**	right-SD < left-SD, controls

Note: Values represent mean percentage correct ± standard deviation. left-SD = left-lateralised semantic dementia; right-SD = right-lateralised semantic dementia; * *p* < 0.05, ** *p* ≤ 0.001.

**Table 3 brainsci-11-01195-t003:** Voxel-based morphometry results showing significant negative correlation between grey matter intensity and fixations to the eyes in all semantic dementia patients and controls combined.

Regions	Hemisphere	MNI Coordinates			Number of Voxels
		x	y	z	
Planum temporale; supramarginal gyrus; precentral gyrus; superior temporal gyrus; Heschl’s gyrus (H1 and H2)	right	62	−22	18	463

Note: Results are voxel-wise and reported at *p* < 0.005 uncorrected for multiple comparisons, with a cluster threshold of 150 voxels. When we included diagnosis as a dummy variable, the same region was significant (peak voxel MNI x = 58; y = −16; z = 18; number of voxels = 29), in addition to two additional clusters in the frontal lobe (right superior frontal gyrus MNI: x = 6, y = 38, z = 54; number of voxels = 49; left superior frontal gyrus MNI: x = −10, y = 40, z = 34; number of voxels = 40).

## Data Availability

Data is available upon reasonable request to the corresponding author.

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
