# Peer review of "Considering Hemispheric Specialization in Emotional Face Processing: An Eye Tracking Study in Left- and Right-Lateralised Semantic Dementia"

_brainsci, 2021, doi:10.3390/brainsci11091195_

Round 1

Reviewer 1 Report

The study by Hutchings et al use eyetracking in clinical cohort of patients with semantic dementia (SD) to investigate the laterality effects of eye fixations on static emotional facial pictures. The study includes 10 patients with dominant left sided SD and 6 patients with right sided SD and 22 healthy controls. Results show greater number of fixations for right sided SD when compared to left sided SD and for fear when compared to controls. Furthermore, the author uses voxel-based morphometry and show negative association of fixations with superior temporal sulcus integrity. The study is well planned, clear in analysis and well written and thus argues for involvement of right hemisphere regions in face processing. The clinical sample size of SD to investigate laterality is perhaps on the conservative side but SD patients are also challenging to recruit. I have a few comments and suggestions listed below which may benefit the manuscript.

  • I would urge the authors to define “facial scanning” in the introduction for the benefit of the readers.
  • Some motivation to use the fear, happy pictures might be warranted. Specifically, as to why not also use sad pictures.
  • “Monocular eye tracking was recorded from the right eye using the EyeLink 1000 (SR 167 Research, Mississauga, Ontario, Canada), with a sampling rate of 500 Hz”. Did authors anticipate any limitations of only using right eye for tracking especially when laterality was to be investigated? Some comments might be needed.
  • Figure3: Can the authors comment on the past literature showing left gaze bias which doesn’t seem to be seen in their results for healthy controls? The heat map seems to suggest that more time was spent looking at left side of the face for healthy controls. Furthermore, in right-SD ventral visual pathway corresponding to left visual field might be affected and for left-SD ventral visual pathway of left hemispheres corresponding to right visual field might be affected. Can the authors comment on how this might be relevant to the results? Some additional analysis to see if this i.e., if fixations of the left half vs right half of the face might be interesting?
  • “Covariate analysis: fixations to the eyes. Next, the relationship between grey matter intensity and number of fixations to the eyes was investigated in all participants combined, averaged across emotions.” I am wondering as to why all the emotions were averaged. Would it better to individually investigate the emotions since averaging might not reveal subtle differences between emotions?
  • “Notably, an fMRI study showed lateralisation to the right hemisphere when processing dynamic faces [64].” There are some differences in the dynamic face which may be more naturalistic than static faces and hence referencing those studies might also require a word of caution and limited interpretation when comparing across studies.
  • Can the authors also remark on the how monocular eyetracking with only right eye might have had an impact on their results? Although similar to comment 3, perhaps some remarks can be made in limitations instead of anticipated shortcomings requested in comment 3.
  • A limitations section will be useful.

Reviewer 2 Report

This study of Hutchings and colleagues aimed to assess the laterality in emotional face processing in sixteen patients with semantic dementia (SD) (10 left-SD, 6 right-SD), compared to 22 healthy control participants. A behavioral facial affect discrimination and facial affect selection task was applied, together with an eye tracking paradigm for which the number of fixations on passively viewed faces expressing different emotions (3 conditions: fear, happy and neutral) was recorded. The author’s main finding was that right-SD showed significantly more fixations to the eyes. Voxel-based morphometry revealed that this was associated with lower volume of the right superior temporal sulcus. The design and research question of the study are interesting and the manuscript is written very well. 

Major points:

  1. One of the research questions was to assess the relationship between number of fixations to the eyes and expression recognition.

Here, I wonder if the acquired eye tracking data could not be explored in more detail, e.g. is there a possibility to perform analyses using pupillometry? It has been shown earlier that pupil responses are associated with emotion processing, in addition to simple autonomic responses to arousing stimuli (e.g. Oliva and Anikin, 2018). This would be an interesting complementary analysis, especially given that the authors conclude that “greater number of fixations to the eyes was beneficial for emotion recognition in right-SD”.

If possible, correlations can be shown as scatter plots.

  1. For voxel-based morphometry, it was mentioned in the methods that “neural correlates of fixation patterns: number of fixations was entered into a GLM that included all participants, to achieve greater variance in scores.”

I personally would not pool controls and patients in GLM models as it appears as one aims to artificially increasing the power to detect an effect. Did the results differ when the GLM was fitted within SD patients? If the authors have good arguments to pool controls and patients: did the GLM include a variable for the dummy coded diagnosis? This is commonly done to avoid bias of one diagnostic group driving the result.

Minor

  1. The initial sample size doesn’t correspond to the sample size included for imaging “Thirty-two scans (20 controls, 7 left-SD, 5 right SD) were available for VBM analysis”.

What was the in/exclusion reason?

  1. In the discussion, it is mentioned that “performance in SD stands in contrast to what we previously observed in bvFTD, where increased fixations to the eyes was not associated with performance on tasks of emotion perception.”

Some studies argue that left and right temporal variants of (TDP-43) FTD should be considered the same disease (e.g. Borghesani et al. 2019). Hence, would the observed effect in right SD be disease-stage dependent, given that a recent study shows that atrophy in left SD (with underlying TDP-43) will spread towards the right hemisphere and vice versa (Borghesani et al. 2019)? What’s the authors opinion on this? This is an important nuance since the discussion also mentions “these preliminary findings.. provide a potential avenue for intervention studies aimed at enhancing emotion processing, particularly in right-SD.”

Round 2

Reviewer 1 Report

I thank the authors for the responses. The authors have edited the manuscript to my satisfaction.